# Investigation on factors affecting early strength of high-performance concrete by Gaussian Process Regression

**Hai-Bang Ly** *, **Thuy-Anh Nguyen, Binh Thai Pham**

University of Transport Technology, Hanoi, Vietnam

* banglh@utt.edu.vn

## Abstract

This study aims to investigate the influence of all the mixture components of high-performance concrete (HPC) on its early compressive strength, ranging from 1 to 14 days. To this purpose, a Gaussian Process Regression (GPR) algorithm was first constructed using a database gathered from the available literature. The database included the contents of cement, blast furnace slag (BFS), fly ash (FA), water, superplasticizer, coarse, fine aggregates, and testing age as input variables to predict the output of the problem, which was the early compressive strength. Several standard statistical criteria, such as the Pearson correlation coefficient, root mean square error and mean absolute error, were used to quantify the performance of the GPR model. To analyze the sensitivity and influence of the HPC mixture components, partial dependence plots analysis was conducted with both one-dimensional and two-dimensional. Firstly, the results showed that the GPR performed well in predicting the early strength of HPC. Second, it was determined that the cement content and testing age of HPC were the most sensitive and significant elements affecting the early strength of HPC, followed by the BFS, water, superplasticizer, FA, fine aggregate, and coarse aggregate contents. To put it simply, this research might assist engineers select the appropriate amount of mixture components in the HPC production process to obtain the necessary early compressive strength.

## 1. Introduction

Concrete is a widely used material in building structures because of its many unique features [1]. Along with the development of construction engineering technologies, the development of new concrete technology also plays a significant role. More and more new materials are being developed, widely applied in modern building structures. High-Performance Concrete (HPC) is one of the new materials with improved physical and mechanical properties, bringing advancements in material technology and construction structure [2, 3]. The concept of HPC is used to describe concrete that is manufactured with high quality, carefully selected raw material ingredients. The optimal proportion of the mixture is then mixed, poured, solidified, and deal with the highest technical standards [1, 4]. Therefore, HPC has outstanding properties

**Data Availability Statement:** All data are available from the UCI database: http://archive.ics.uci.edu/ml/datasets.php.

**Funding:** The author(s) received no specific funding for this work.

**Competing interests:** The authors have declared that no competing interests exist.

compared to conventional concrete such as high compression strength, very high tensile strength, high elastic modulus, sustainability and stability under adverse impacts of aggressive erosion of environment, and many other characteristics, very useful in building transport infrastructure [1, 3, 5]. As a result, HPC is increasingly used for highway applications, including new construction, repair and rehabilitation, bridges, tunnels, and high-rise buildings [6]. HPC requires tighter materials than concrete that is usually specified by ASTM standards. Concrete Portland Cement (PC), water, fine sand and coarse aggregate (fine and coarse aggregate) are the fundamental components of HPC, although there are other cement ingredients including fly ash, blast furnace slag, and chemical additives (i.e., superplasticizer [7]). It is vital to emphasize the long-term profitability of the cement sector since the additional cement materials contribute to reducing the quantity of $CO_2$ released throughout the cement production process. Furthermore, since the majority of these compounds are by-products of industrial activities, reusing them is beneficial. As a result, it is predicted that HPC would grow more popular in the next decades, mostly due to its high levels of sustainability and durability. However, the addition of these components in HPC makes calculating the HPC mixing ratio and HPC behavior model significantly more complicated than similar processes for conventional cement [1]. The HPC mixed design method was proposed by ACI [8], Aitcin [5], Islam Laskar and Talukdar [9] with the purpose of obtaining a combination of component materials and proportions respectively to create HPC with improved properties. The compressive strength of the twenty-eighth day is the most widely used target function in mixed designs. Some studies show that the compressive strength of HPC is influenced by many factors such as aggregate, cement, blast furnace slag, fly ash, the content of water, superplasticizer, and testing age [5, 10–12]. However, waiting 28 days to obtain compressive strength for 28 days is time-consuming, and in some cases, the specified intensity needs to be reached at an earlier age to speed up construction. Thus, predicting the concrete compressive strength at an early age is an active research area in civil engineering, which will facilitate construction and restoration tasks to improve quality [6]. At the same time, the more achieved information about the relationship of concrete composition versus strength is, the better explored the nature of concrete is. Thus, the optimization of the concrete mixture allows adjusting the mixing ratio to avoid concrete not reaching the required compressive strength, saving time and construction costs [3, 13].

In recent years, machine learning (ML) has been exploited to be a powerful numerical tool for solving many real-world problems, especially in civil engineering. This approach can discover a complicated relationship between the inputs and output for better accuracy in an analysis using different computational algorithms. Various ML-based models have been introduced and applied for different purposes like the prediction of landslides [14], floods [15], prediction of mechanical properties of materials [16–18], and structures [19, 20]. In general, the results of the studies demonstrate the potential of ML-based models in analyzing and modeling complex problems, which can be applied to evaluate the effect of factors on the early strength of HPC. For example, the studies of Yeh et al. [7, 21, 22] built an ANN model with a backpropagation algorithm to predict the compressive strength of concrete at different ages (3 days, 14 days, 28 days, and 90 days). In another study, Melda Yucel and Ersin Namli [23] used eight different ML and Extreme learning machines (EML) techniques drafted from Random Forest (RF) and ANN algorithms to compare the predictive compressive performance of HPC. Mustapha and Mohamed [1] proposed a Support Vector Machine (SVM) model that performed well in forecasting concrete's compressive strength. However, these works showed the relationship between concrete components that influence the compressive strength without establishing the relevance of each kind of material that affects compressive strength.

In summary, understanding the parameters that influence the early strength of concrete in general, and high-performance concrete in particular, is critical throughout the design process.

The early strength of HPC is also a crucial component to consider, and it should be carefully managed throughout the earliest stages of the construction phase. Furthermore, the early strength is critical since it also dictates the age at which formwork may be removed. Thus, in this work, the main objective is to use one of the most popular ML-based models, namely Gaussian Process Regression (GPR), to predict and evaluate the effect of factors on the early strength of HPC. Taking advantage of the well-known concrete database in the literature, the experimental results of 324 HPC samples were selected and used to generate the training and testing datasets for the construction and validation of the ML algorithm. The inputs considered in this work were cement, blast furnace slag, fly ash, the content of water, superplasticizer, coarse aggregate, fine aggregate, and the testing age, whereas the early strength of HPC was the only output of the problem. It is worth mentioning that other cement replacement materials, such as silica fume or metakaolin, were not included in the current database. The analysis of these variables could be the aim of another study in the near future. Pearson correlation coefficient (R), root mean square error (RMSE), and mean absolute error (MAE) were used to evaluate the performance of the model.

## 2. Data collection and preparation

A dataset containing 324 early strength HPC samples from previous works of Yeh [7, 21, 22] was used in this work. The HPC samples were obtained using the original inputs, including cement, blast furnace slag (BFS), fly ash (FA), content of water, superplasticizer, coarse aggregate (Coarse Agg.), fine aggregate (Fine Agg.) and the testing age.

The data were split into the training part (70% data) and the testing part (30% the remaining data). It is worth noticing that all inputs cover a wide range of values. For instance, the cement content was in the range of [102, 540] (kg), the BFS content was in the range of [0, 359.4] (kg), the FA content was in the range of [0, 174.7] (kg), the water content was in the range of [121.8, 228] (kg), the superplasticizer content was in the range of [0, 32.2] (kg), the coarse aggregate content was in the range of [822, 1134.3] (kg), the fine aggregate content was in the range of [594, 992.6] (kg), and the testing age was in the range of 1 to 14 days. The target of the study, the early compressive strength, ranged in the [2.33, 59.76] (MPa). It is worth noticing that the given values of input variables corresponded to $1m^3$ of the concrete mixture. Primarily, statistical analysis was conducted, which revealed that there was no substantial cross-correlation in the input space. As a result, machine learning models may be trained with a high degree of generalization capacity. Fig 1 displays the histograms of all variables in this work.

## 3. Gaussian Process Regression (GPR)

Various successful applications of machine learning algorithms in materials science have been presented—for instance, Kernel Ridge regression [24], Recursive Neural Networks [25], Artificial Neural Network [26], Radial Basis Function Neural Network [27]. In the work, Gaussian Process Regression (GPR) is used to predict the early compressive strength of HPC, aging from 1 to 14 days. In general, a Gaussian Process Regression could be understood as the generalization of the Gaussian probability distribution prior to interpolation by a Gaussian process using covariances. This algorithm is a nonparametric and Bayesian approach usually used to treat problems related to nonlinear regression [28, 29], and classification [30]. The main idea of GPR lies in the learning process of such an algorithm. Many supervised ML algorithms learn exact values from the dataset, whereas GPR infers a probability distribution over the values of the dataset. GPR estimates the probability distribution of all admissible functions that

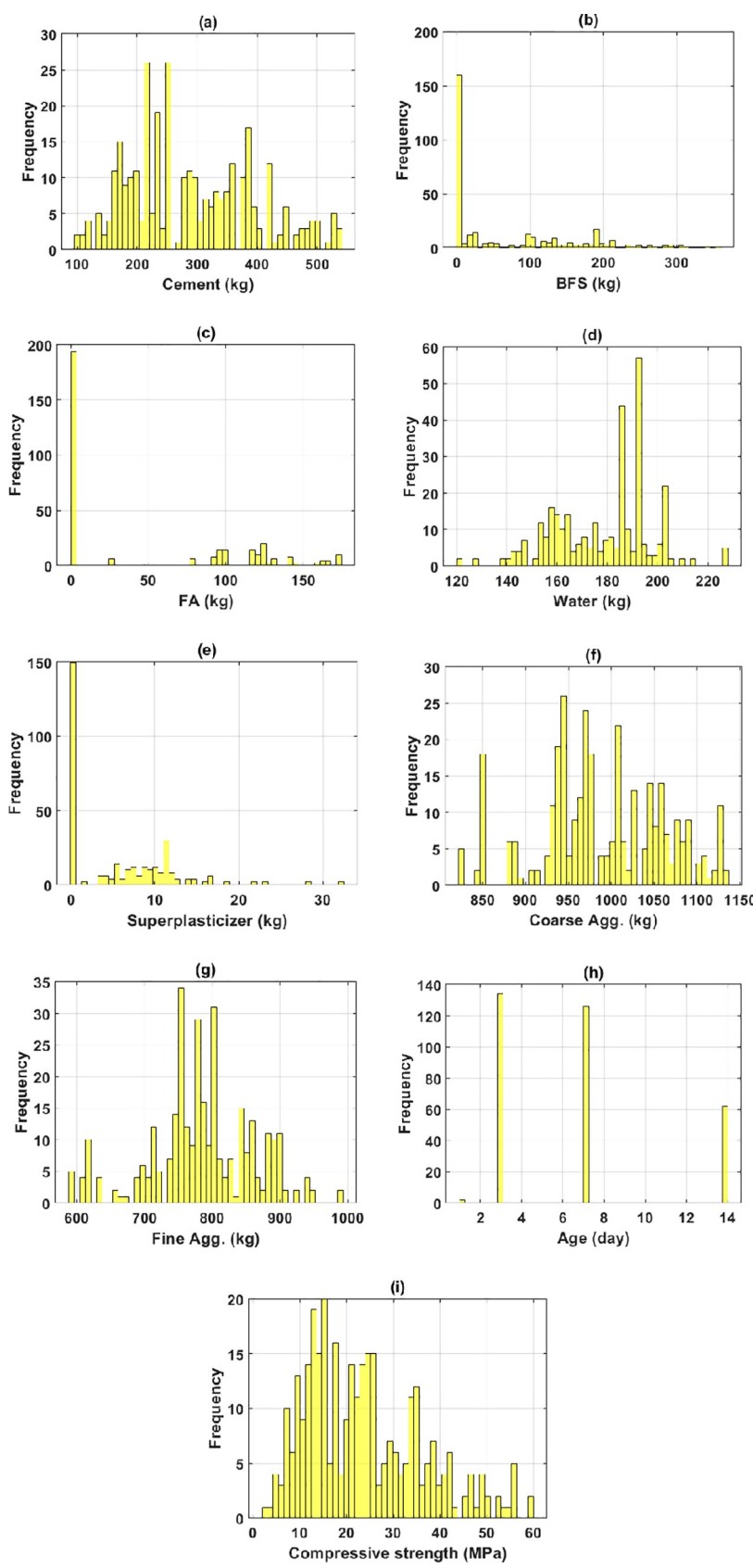

**Fig 1.** Histograms of the inputs in the current database, (a) cement; (b) blast furnace slag; (c) fly ash; (d) water; (e) superplasticizer; (f) coarse aggregate; (g) fine aggregate; (h) age; (i) early compressive strength.

could reasonably fit the data space regarding the regression problems. For GPR development and all possible algorithm applications, the readers could be referred to the literature [31, 32].

In this work, the GPR model was trained to take advantage of the GPR implementation in Matlab routine but adapted to the problem. The basis function in the Matlab routine contains many possibilities, i.e., none, constant, linear, pure-quadratic, or function handle. After an extensive trial-and-error test, the constant function was adopted for the basis function. The adopted Kernel function was selected as "squared exponential" by performing similar preliminary tests. Besides, the values of Sigma, reflecting the initial value for the standard noise deviation of the model, were chosen with the help of the hyper-parameter optimization function in the Matlab routine. Last but not least, the Mean Squared Error was chosen as the cost function, whereas the k-fold cross-validation was chosen as k = 10.

In this study, the accuracy of GPR was evaluated by commonly used statistical criteria, for instance, the Pearson correlation coefficient (R), root mean squared error (RMSE), and mean absolute error (MAE). These criteria are important parameters in regression analysis, which interpret the relationships between the predicted and actual outputs in different ways [33, 34]. For example, a higher value of R shows a good correlation, whereas lower values of RMSE and MAE measure the error between them and indicate the algorithm's better performance. These criteria can be expressed as:

$$R = \sqrt{1 - \frac{\sum_{i=1}^{M}(x_i - y_i)}{\sum_{i=1}^{M}(x_i - \bar{y})}} \tag{1}$$

$$RMSE = \sqrt{\frac{1}{M}\sum_{i=1}^{M}(x_i - y_i)^2} \tag{2}$$

$$MAE = \frac{1}{M}\sum_{i=1}^{M}|(x_i - y_i)| \tag{3}$$

where $M$ is the number of data; $x_i$ and $y_i$ are the actual and predicted outputs, respectively; and $\bar{y}$ is the mean of the predicted output.

## 4. Results and discussion

### 4.1. Construction and validation of GPR black-box

In this section, the development of the GPR model is performed, mainly based on the performance of GPR in predicting the early compressive strength of HPC. In order to obtain reliable prediction results, two parameters were varied in the simulation: the ratio of samples between the training and testing datasets and the sampling technique used for the selection of samples. Generally, the train/test ratio is taken as 70/30, as recommended in many studies. However, different train-to-test ratios were adopted in several works, especially in the contribution of Yeh [7, 21, 22]. Therefore, the three most chosen train-to-test ratios were selected in this work to investigate the influence of the training dataset size on the prediction accuracy of the

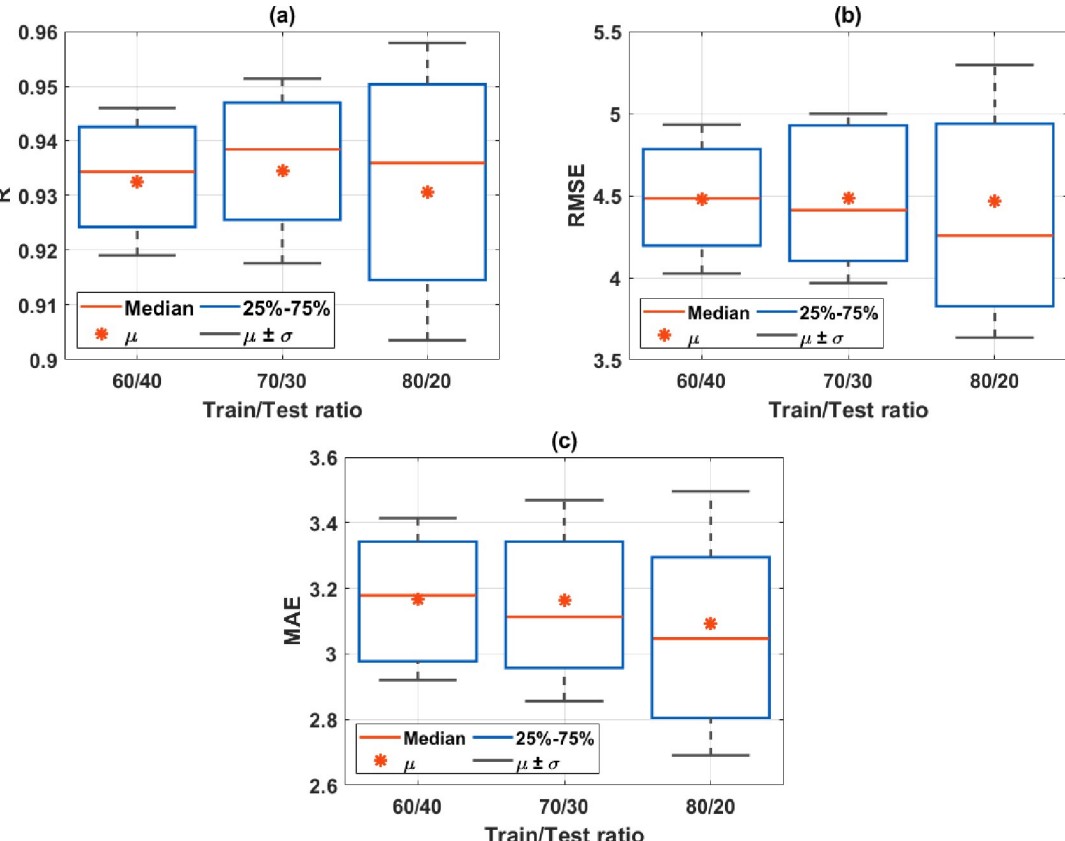

**Fig 2.** Summary of the prediction performance over 100 simulations using different train-to-test ratios and different criteria: (a) R; (b) RMSE; and (c) MAE, where μ denotes the average value, σ denotes the standard deviation, and 25%-75% denotes the values in the range of the first and the third quartiles, respectively.

machine learning model. Moreover, 100 simulations were performed in each case, taking into account the random sampling effect, as it was proven to affect the prediction results of machine learning algorithms significantly. Fig 2 displays the summary of the prediction performance over 100 simulations in function of different training/testing ratios and the random sampling effect. It can be seen that the variation in function of R, RMSE, and MAE was low with the 60/40 train-to-test ratio. However, in terms of the median (denoted as Median) and average values (denoted as μ), the 60/40 ratio exhibited the lowest prediction accuracy.

On the contrary, the lowest values of RMSE and MAE were obtained in the case of the 80/20 train-to-test ratio, showing the best prediction performance. However, the variation of the two quartile levels (first and third) and standard deviation (σ) were significant. Finally, the 70/30 train-to-test ratio was taken because such a ratio exhibited the best prediction accuracy regarding R, along with a similar variation level of quartiles and standard deviation compared with the case of the 60/40 train-to-test ratio.

Over 100 simulations in the case of the 70/30 training/testing datasets ratio, one GPR model was chosen to perform the PDP analysis of the factors affecting the early compressive strength of HPC. The selection was based on the three performance criteria representing the prediction accuracy of the models. A selected model exhibited the highest value of R for the testing part, and the lowest values of RMSE and MAE, for the testing dataset. Fig 3 displays the predicted early compressive strength versus the corresponding targets of the selected GPR model associated with the training and testing parts. The fitting linear lines (discontinuous

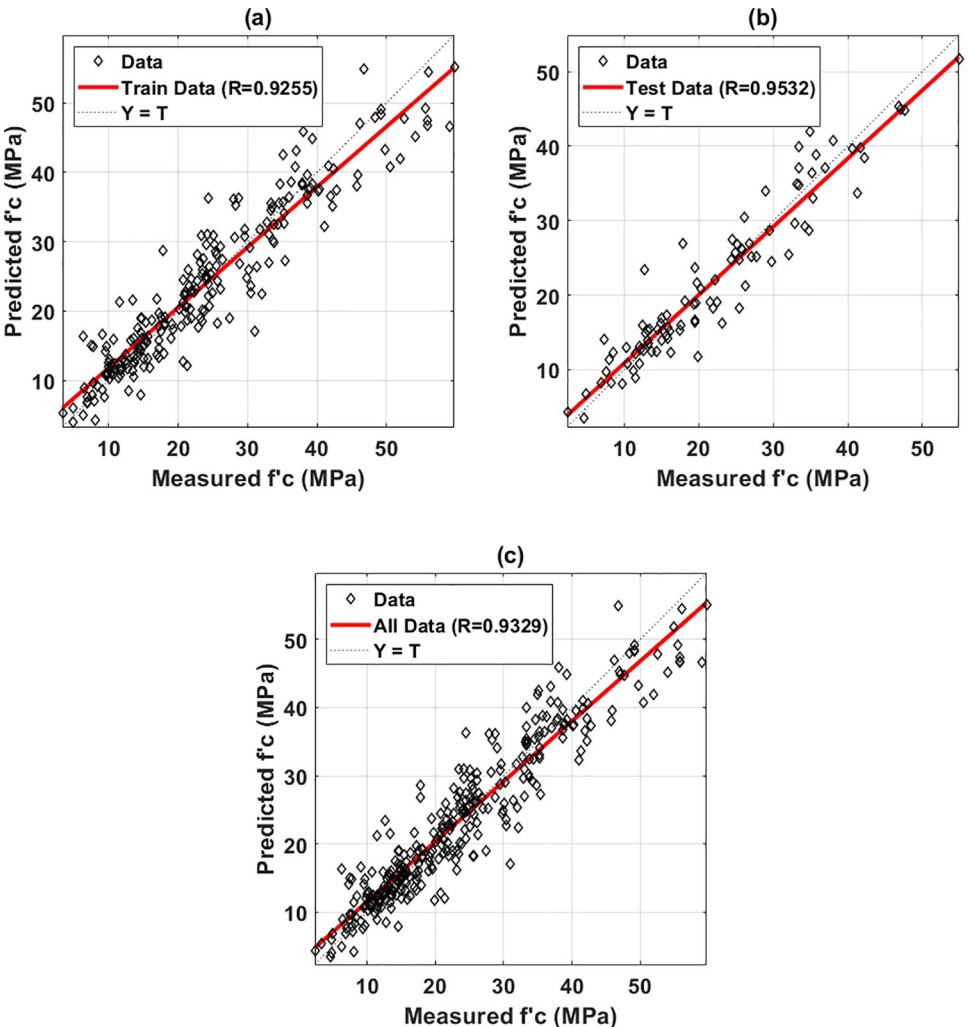

**Fig 3.** Regression graphs for the measured and predicted values of early compressive strength of HPC for (a) the training dataset; (b) testing dataset; and (c) all dataset.

black lines) were also plotted in each case to demonstrate the GPR model's performance. Table 1 illustrates the various error criteria used to compare predicted and experimental values of early compressive strength for both training and testing sets. High correlation values were obtained for both the training dataset (R = 0.9255) and the testing dataset (R = 0.9532). Besides, the values of standard deviation error were 4.8703 and 3.3790 for the training, and testing dataset, respectively. Thus, it can be seen that the performance of the testing dataset was superior to the training one, which might come from the samples that constituted these

**Table 1. Summary of the statistical measures for the training and testing datasets.**

|  | MAE | RMSE | Error Mean | Error St.D. | R |
|---|---|---|---|---|---|
| Training part | 3.3425 | 4.8598 | 0.0493 | 4.8703 | 0.9255 |
| Testing part | 2.5286 | 3.3630 | -0.1216800 | 4.1398 | 0.9532 |

St.D. = Standard deviation.

parts. It is worth noticing that in this study, the classical random sampling effect was performed to select the sample indexes for the training part (70% of the total data) and the testing part (30% of the remaining data). This might be the main reason that influenced the accuracy of the GP algorithm. However, with the 70/30 train-to-test ratio, as previously shown, the variation of R was slight and in an excellent range of accuracy (i.e., about 0.92 to 0.95). Therefore, it could be concluded that the results in this study exhibited high credibility, and the developed GPR model could be used for further investigation.

## 4.2. Comparison with literature

In order to demonstrate the performance of the constructed GPR model, comparisons with existing results in the literature are conducted in this section. Table 2 summarizes the previous works in HPC compressive strength prediction using machine learning algorithms. The highlight of previous studies included the reference, the machine learning algorithm used, the concrete content used, the sample size, and the quality metric. Various machine learning methods have been employed, such as Support Vector Machine (SVM), Genetic operation trees (GOT), Artificial Neural Network (ANN), Neural-fuzzy inference system (NFIS), Bagging regression trees (Bagged), Fuzzy polynomial neural networks (FPNN), etc. In terms of concrete content, main contents such as cement (C), silica fume (SF), water (W), fine (F.Agg) and coarse aggregates (C.Agg), and superplasticizer (SP) have been used. In addition, silica (S) as a binder has been used in Kasperkiewicz and Dubrawski [35], Raghu Prasad et al. [36], and Fazel Zarandi et al. [37]. Blast-furnace slag (BFS) has been used in Yeh et al. [7], Yeh and Lien [38], Deepa et al. [39], Chou Jui-Sheng et al. [40]. Fly ash (FA), as a cement replacement, has been employed in Yeh et al. [7], Yeh and Lien [38], Raghu Prasad et al. [36], Deepa et al. [39], Chou

**Table 2. Comparison with literature for prediction of compressive strength of HPC.**

| Ref. | Machine learning algorithm | Concrete content | Sample size | Values of R |
|---|---|---|---|---|
| Kasperkie wicz and Dubrawski [35] | Fuzzy-adaptive resonance theory-MAP ANNs | C, S, SP, W, F.Agg, C.Agg | 340 | 0.7842 |
| Yeh et al. [7] | ANN | C, FA, BFS, W, SP, C.Agg, F.Agg | 727 | 0.9560 |
| Yeh and Lien [38] | GOT, ANN | C, FA, BFS, W, SP, C.Agg, F.Agg | 1196 | 0.9311 |
| | | | | 0.9663 |
| Raghu Prasad et al. [36] | ANN | C, W, FA, microsilica, C.Agg, F.Agg | 24 | 0.9165 |
| Hoang et al. [41] | Least-Square SVM, ANN | C, fine aggregate, small coarse aggregate, medium- coarse aggregate, W, SP | 239 | 0.9327 |
| | | | | 0.90 |
| Deepa et al. [39] | MLP, Linear regression, M5P model tree | C, BFS, fly ash, W, SP, C.Agg, F.Agg | 300 | 0.7908 |
| | | | | 0.7009 |
| | | | | 0.8872 |
| Chou Jui-Sheng et al. [40] | ANN, Multiple regression, SVM, Bagged | C, FA, BFS, W, SP, C.Agg, F.Agg | 1030 | 0.9535 |
| | | | | 0.7818 |
| | | | | 0.9412 |
| | | | | 0.9436 |
| Rajiv Rupta et al. [44] | NFIS | C, W, C.Agg, F.Agg, average slump | 864 | 0.8718 |
| Pham Anh-Duc et al. [42] | ANN, SVM, Least Square SVM | C, C.Agg, F.Agg, medium coarse aggregate, W, SP | 239 | 0.90 |
| | | | | 0.9110 |
| | | | | 0.9434 |
| Fazel Zarandi et al. [37] | FPNN | C.Agg, F.Agg, SP, SF, W, and C | 458 | 0.9060 |
| This work | GPR | C, BFS, FA, W, SP, C.Agg, F.Agg | 324 | 0.9522 |

Jui-Sheng et al. [40] and this work. Besides, Hoang et al. [41] and Pham Anh-Duc et al. [42] have used three ranges of dimensions for classifying aggregate size, such as fine, medium, and coarse. Regarding the assessment of the prediction performance, notably, R varied from 0.7009 to 0.9663 and could be considered an essential deviation in predicting the compressive strength of HPC. The investigation of this work achieved excellent results for R (R = 0.9522), close to the results reported for Artificial Neural Network. Unlike previous works, this study employed k-fold cross-validation, as indicated previously. Such cross-validation allowed ensuring good generalization capability [43]. The present work also standardized the work in this field by exploring hidden nonlinear complex relationships between concrete contents and compressive strength of HPC through ICE and PDP analyses.

The predictive performance of GPR was compared with two state-of-the-art machine learning algorithms, namely ANN and SVM. The same training dataset was used to train the two models, whereas the predictive performance was evaluated using the same testing dataset. Regarding the ANN model, one single hidden layer structure was adopted for comparison, with 9 neurons in such hidden layer. The training algorithm was selected as the default training function in Matlab programming language (i.e., Levenberg-Marquardt algorithm), whereas the "tansig" transfer function for the hidden layer and linear function for the output layer, were adopted for comparison. With respect to SVM model, Bayesian optimization was used to find the appropriate hyperparameters for SVM, the "fitrsvm" function in Matlab. The final hyperparameters adopted for comparison were BoxConstraint = 19.05, Epsilon = 13.49, Kernel function = polynomial, Kernel polynomial order = 3, Kernel scale = auto, solver = Sequential minimal optimization (SMO). The prediction results are shown in function of all dataset (Fig 4). It can be seen that although high accuracy was achieved (i.e., $R_{ANN}$ = 0.9274, $R_{SVM}$ = 0.9025), the prediction performance of these two models was slightly inferior to the proposed GPR model in this study ($R_{GRP}$ = 0.9329).

Although the GPR model outperformed the ANN and SVM models in predicting the compressive strength of HPC, it is important to note that each machine learning technique has its own set of benefits and drawbacks that must be considered. First and foremost, the computational requirements for GPR are much higher than those for ANN. As a result, this model can only be employed with small datasets. Furthermore, GPR is better than ANN in dealing with missing data and can properly collect hidden information, even in regions with few available

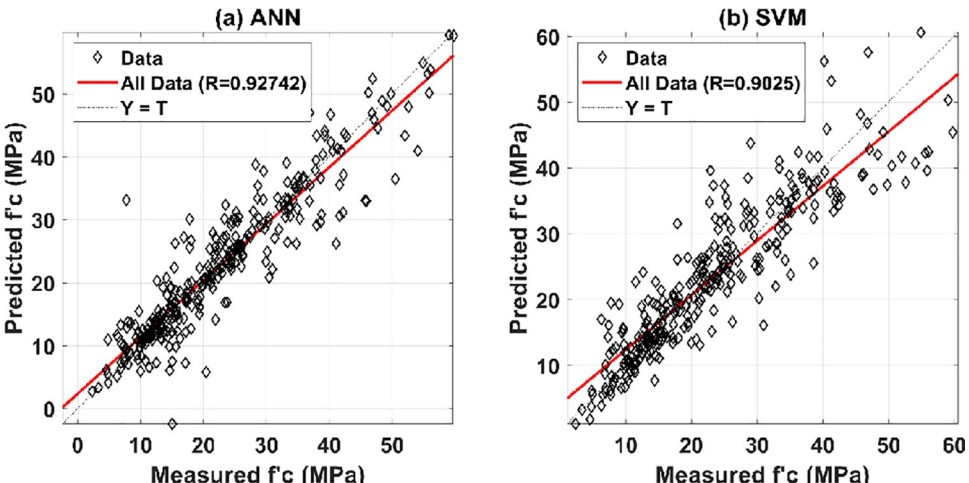

**Fig 4.** Regression graphs for the measured and predicted values of early compressive strength of HPC for all dataset: (a) ANN; and (b) SVM.

training data points. This is GPR's primary advantage over other machine learning algorithms and the reason for the PDP analysis in the following sections. Additionally, many different kernels are available for use in the GPR model (standard kernels are supplied in Matlab, for instance), but it is also possible to custom the Kernels. Finally, this method makes the prediction using the whole set of features information. This implies that GPR may become inefficient when the number of features exceeds a few dozens.

## 4.3. Interpretation of results using Partial Dependence Plots

Once the GP black-box was constructed and validated, it was used to perform Partial Dependence Plots (PDP) analysis. Such a method is widely adopted to analyze the influence of the input variable on the predicted output in machine learning problems [45]. Besides, Individual Conditional Expectation (ICE) plots displayed by each line show the results obtained by changing every instance and seeking the variation of the predicted output with that change [46]. Thus, a PDP analysis is obtained by computing the average of the lines obtained by ICE plots.

Fig 5 shows the PDP analysis between cement, BFS, FA, superplasticizer, and testing age with the early compressive strength of HPC. These input parameters were observed, influencing the output positively. First, it can be seen that the cement content was the most critical input variable, as the early compressive strength varied from 10.28 to 42.24 MPa within the range of cement content. Similar findings on the early strength of concrete have also been reported in the literature [47]. The testing age was the second important input variable (i.e., early compressive strength varied from 14.42 to 34.37 MPa), followed by the superplasticizer content (i.e., early compressive strength varied from 21.06 to 32.12 MPa), BFS content (i.e., early compressive strength varied from 21.81 to 31.10 MPa), and FA content (i.e., early compressive strength varied from 23.76 to 26.16 MPa). The PDP curves of compressive strength were almost in a linear relationship with cement, FA, and BFS. An exponential relationship was preferred to relate the superplasticizer content and early compressive strength of HPC, whereas a logarithmic equation could be fitted between the testing age and the predicted target.

Second, the water content, coarse and fine aggregates were the factors that exhibited a negative effect on the early compressive strength of HPC (Fig 6). The water content exhibited the most significant adverse effect on the predicted output (i.e., early compressive strength varied from 31.23 to 15.48 MPa). The aggregates' content, both coarse and fine, seemed to have a lower influence on the early compressive strength, as summarized in Table 3. Precisely, the early compressive strength varied from 26.17 to 20.17 MPa, and from 26.86 to 20.38 MPa in the range of content of coarse and fine aggregates, respectively. Even though a negative effect on the average curves of PDP was observed for both fine and coarse aggregates, this observation still needs further investigation. From ICE plots, it can be seen that the aggregates exhibited a positive effect on about 20 to 30% of the samples (see S1 Appendix for better illustration). Therefore, gathering more available data in the literature could help explore the effect of these two types of aggregates on the early compressive strength of HPC.

PDP analysis shows advantages in analyzing the influence of input variables, especially in combining the effect of two input variables. As the cement content was classified as the most critical feature, several relationships with other input variables were investigated in order to highlight the coupling effect of 2 parameters to the early strength of HPC.

As can be seen in Fig 7, the maximum early compressive strength of HPC could be obtained with a high content of cement with almost any FA, superplasticizer, coarse aggregate contents. This observation also confirmed that the contents of FA, SP, and coarse aggregates would not

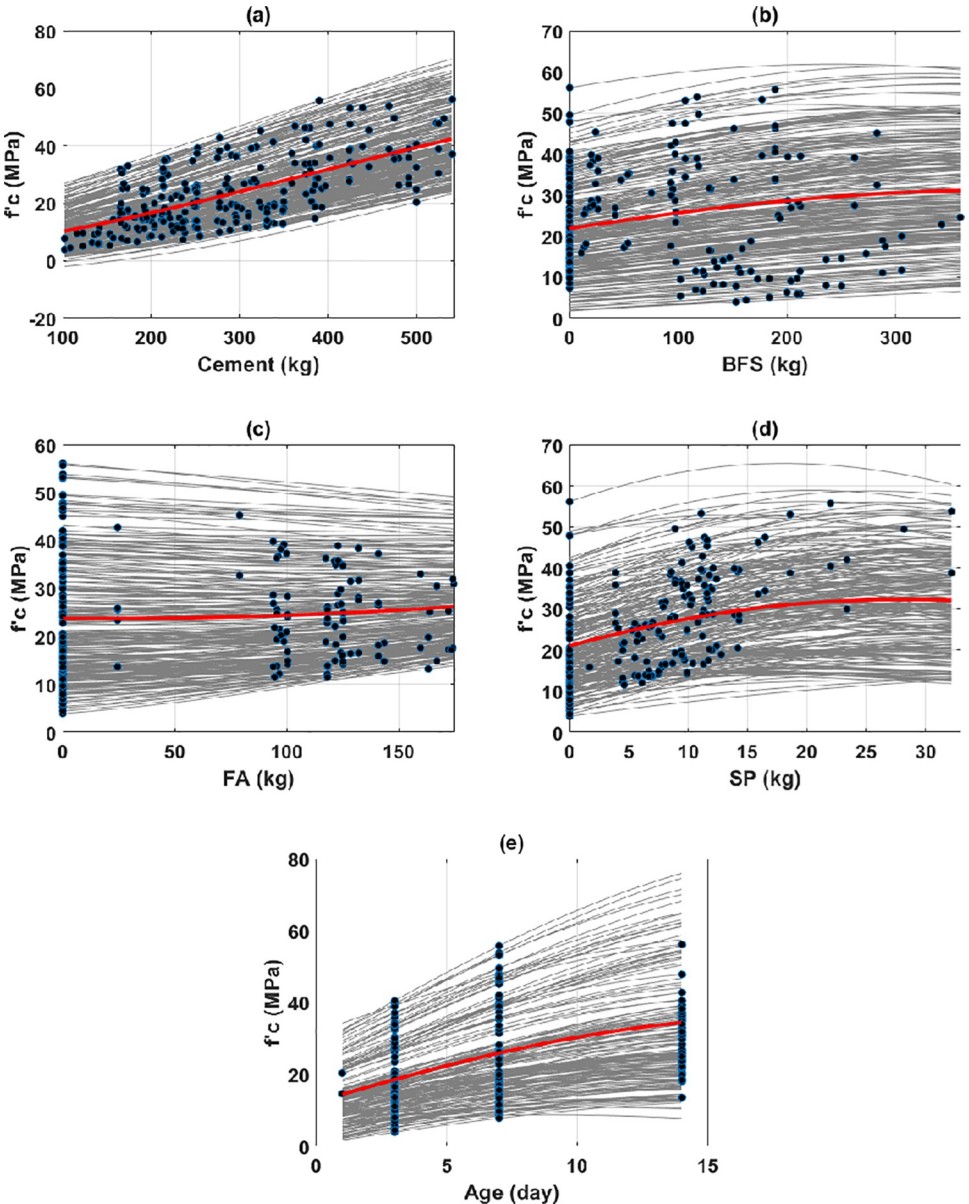

**Fig 5.** ICE and PDP curves in function of input variables for: (a) cement; (b) blast furnace slag; (c) fly ash; (d) superplasticizer; and (e) age.

significantly affect the early strength of HPC. Generally, the role of FA or silica fume is more pronounced at the later age of concrete. Indeed, the advantage of HPC is that it often incorporates pozzolanic or latent hydraulic additional components, such as fly ash, silica fume, and GGBS. These elements react with (or are activated by) alkali created by the hydration of cement to form specific compounds that improve the strength. The basic benefit of fly ash is its reactivity with the available lime and alkali in concrete, creating more and more cementitious compounds over time. The pozzolanic reaction of fly ash with lime gives an additional calcium silicate hydrate binder (C-S-H). Besides, silica fume, with its fineness at approximately two orders of magnitude finer than the others, has an extremely large surface area. Because silica fume is so reactive, it consumes the alkali that the cement eventually releases. Besides, the

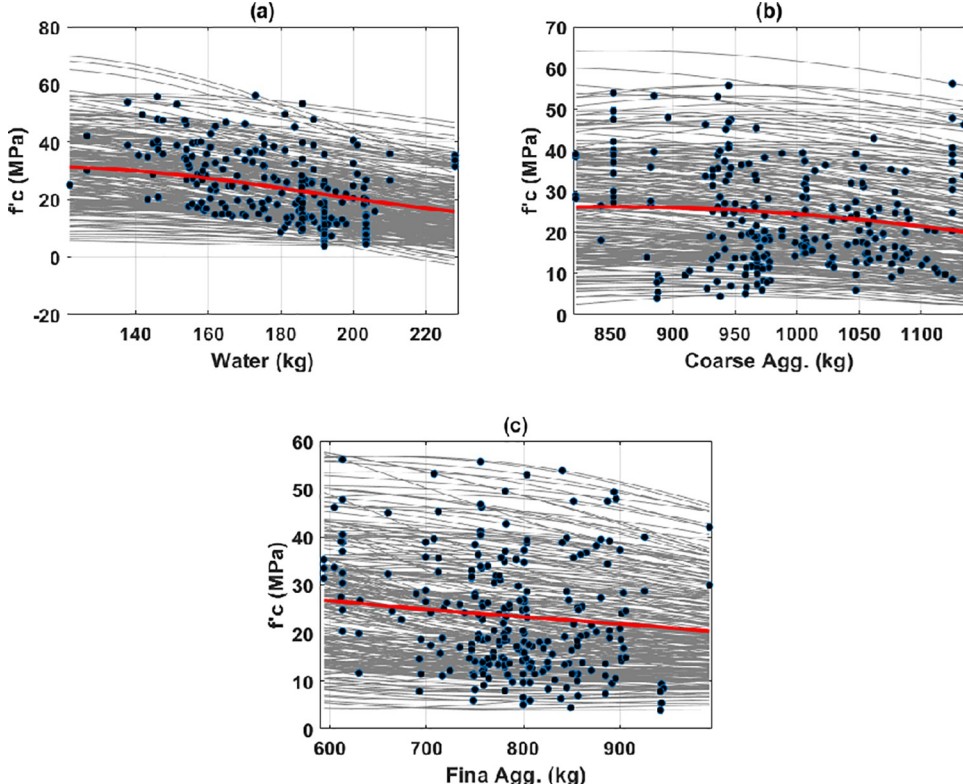

**Fig 6.** ICE and PDP curves in function of input variables for (a) water; (b) coarse aggregates; and (c) fine aggregates.

findings in this study are also in good agreement with the literature [48]. Differently, the BFS content was required at least 100 kg, and the testing age was superior to 7 days to achieve high early compressive strength (i.e., superior to 30 MPa). The water content exceeded 170 kg, or the fine aggregate content exceeded 800kg would decrease the early compressive strength of HPC.

Overall, the two-dimensional PDP curves could be helpful for engineers in designing the appropriate content of all the constituents of HPC. However, it is worth noticing that this

**Table 3. PDP investigation of the compressive strength in function of different inputs and the corresponding effects, rank.**

| Inputs | Input variation | | PDP Compressive strength variation | | | Effect | Rank |
|---|---|---|---|---|---|---|---|
| | Min | Max | Min | Max | \|Δ\| | | |
| Cement | 102 | 540 | 10.28 | 42.24 | 31.96 | Positive | 1 |
| BFS | 0 | 359.4 | 21.81 | 31.10 | 9.29 | Positive | 5 |
| FA | 0 | 174.7 | 23.76 | 26.16 | 2.4 | Positive | 8 |
| Water | 121.8 | 228 | 15.75 | 31.23 | 15.48 | Negative | 3 |
| Superplasticizer | 0 | 32.2 | 21.06 | 32.12 | 11.06 | Positive | 4 |
| Coarse Agg. | 822 | 1134 | 20.17 | 26.17 | 6.0 | Negative | 7 |
| Fine Agg. | 594 | 945 | 20.38 | 26.86 | 6.48 | Negative | 6 |
| Age | 1 | 14 | 14.42 | 34.37 | 19.95 | Positive | 2 |

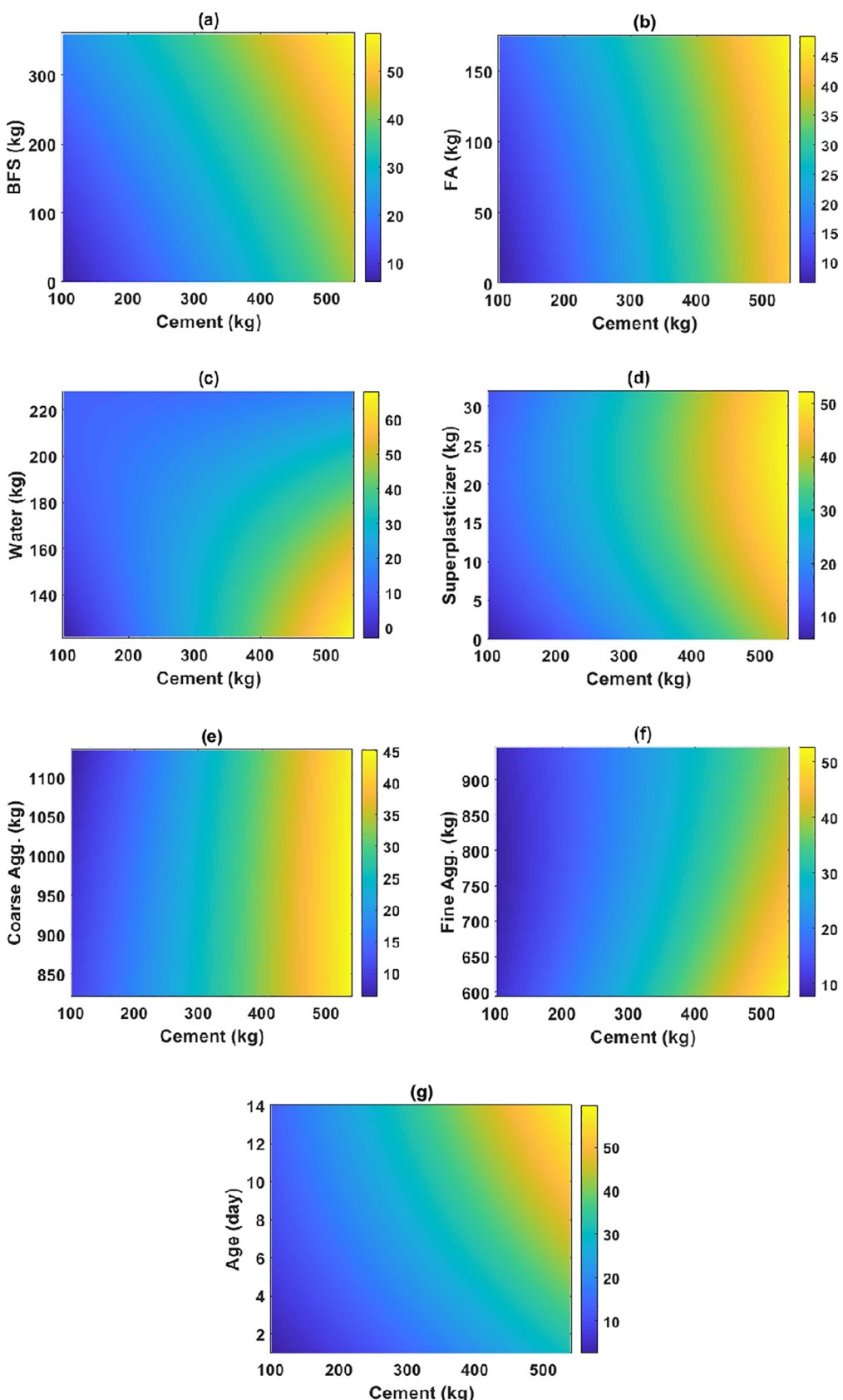

**Fig 7.** Two-dimensional PDP curves analysis between cement and other input variables for: (a) BFS; (b) FA; (c) water; (d) superplasticizer; (e) coarse aggregates; (f) fine aggregates; and (g) age. The color scale presents a variation of compressive strength in MPa.

study's findings need further improvement, as more data should be collected to cover a broader range of input variables.

## 5. Conclusion

This study investigates the relationships between the HPC constituents and the corresponding early compressive strength, using Gaussian Process Regression as a machine learning-based algorithm. A dataset containing 324 experiments on HPC was used to generate the training and testing datasets for developing the GPR algorithm. The considered HPC ingredients were cement, BFS, FA, water, superplasticizer, coarse and fine aggregates, and the early testing age of HPC. In addition, the prediction capability of the GPR model was evaluated using three well-known statistical measurements, such as RMSE, R, and MAE.

The results showed that the GPR algorithm was a good predictor in predicting the HPC early compressive strength, with R = 0.9532, RMSE = 3.3630 (MPa), and MAE = 2.5286 (MPa) for the testing dataset. The best GPR model was carefully selected after performing the prediction with different sub-datasets taking into account the random sampling effect to construct the training part. Once constructed and validated, a one-dimensional PDP analysis was performed and showed that the content of cement was the most sensitive and vital factor to the early strength of HPC, followed by the testing age, water content, superplasticizer, BFS, FA content, fine aggregate, and coarse aggregate. In addition, two-dimensional PDP analysis revealed many concrete constituent situations to achieve the desired values of early strength of HPC.

In general, this study may be valuable in aiding engineers in calculating the appropriate content of mixture components to utilize in the manufacturing process of HPC. Not to mention, cement replacement materials such as silica fume or metakaolin have been shown to be successful in substituting cement while also significantly increasing the mechanical properties of HPC. This has the potential to be a flourishing research topic with applications in a variety of fields if properly developed. Due to the fact that these two components were not included in this study, the primary focus of future research will be to expand the present database and analyze the influence of these factors on the mechanical behavior of HPC.

## Supporting information

**S1 Appendix.**
(DOCX)

## Author Contributions

**Conceptualization:** Hai-Bang Ly, Thuy-Anh Nguyen, Binh Thai Pham.

**Data curation:** Thuy-Anh Nguyen.

**Formal analysis:** Hai-Bang Ly, Thuy-Anh Nguyen.

**Investigation:** Hai-Bang Ly, Binh Thai Pham.

**Methodology:** Hai-Bang Ly.

**Project administration:** Hai-Bang Ly.

**Software:** Binh Thai Pham.

**Validation:** Hai-Bang Ly, Thuy-Anh Nguyen, Binh Thai Pham.

**Writing – original draft:** Hai-Bang Ly, Thuy-Anh Nguyen.

**Writing – review & editing:** Hai-Bang Ly, Thuy-Anh Nguyen, Binh Thai Pham.

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
