## [Decision Letter · Decision Letter 0]

11 Oct 2021

PONE-D-21-25655Investigation on Factors Affecting Early Strength of High-Performance Concrete by Gaussian Process RegressionPLOS ONE

Dear Dr. LY,

Thank you for submitting your manuscript to PLOS ONE. After careful consideration, we feel that it has merit but does not fully meet PLOS ONE’s publication criteria as it currently stands. Therefore, we invite you to submit a revised version of the manuscript that addresses the points raised during the review process.

We look forward to receiving your revised manuscript.

Kind regards,

Tianyu Xie, Ph.D.

Academic Editor

PLOS ONE

Journal Requirements:

Reviewers' comments:

Reviewer's Responses to Questions

**Comments to the Author**

1. Is the manuscript technically sound, and do the data support the conclusions?

Reviewer #1: Yes

Reviewer #2: Yes

2. Has the statistical analysis been performed appropriately and rigorously? 

Reviewer #1: Yes

Reviewer #2: Yes

3. Have the authors made all data underlying the findings in their manuscript fully available?

Reviewer #1: Yes

Reviewer #2: Yes

4. Is the manuscript presented in an intelligible fashion and written in standard English?

Reviewer #1: Yes

Reviewer #2: Yes

5. Review Comments to the Author

Reviewer #1: This paper integrates the Gaussian Process Regression (GPR) algorithm for the prediction of early compressive strength of high performance concrete (HPC) using a database including 324 HPC samples showing the influence of input mixture variables on the early strength of HPC. This paper looks interesting and well-written. However, this reviewer recommends the publication of this paper after the following concerns are successfully addressed.

1. There should be more discussion on the effectiveness and applicability of HPC concrete compared to traditional concrete in the Introduction. Please discuss the sustainability of high performance concrete (HPC) in the introduction as it includes industrial wastes such as fly ash and slag making it a greener concrete.

2. Please rephrase “However, in these studies, it was only determined in determining the proportion of material components affecting the compressive strength of concrete without showing the importance of each type of material affecting the compressive strength of HPC”.

3. There should be more discussion on the advantages (e.g. efficiency with small datasets) and disadvantages (e.g. inefficiency in high dimensional spaces) of Gaussian Process Regression (GPR) algorithm compared to other machine learning algorithms in Section 3.

4. As shown in Figure 3 and Table 1, testing set has a better predictive performance (R = 0.9532) than training set (R = 0.9255) which is not common for machine learning methods where training set has generally a better predictive performance than testing set. Please explain why it happens.

5. In section 4.2, the effectiveness of GPR is shown through a comparison with other machine learning algorithms in the literature with different sample sizes and input variables. However as the predictive performance of machine learning algorithms depends on the input variables and sample size, please show the predictive effectiveness of the presented method compared to another machine learning algorithm (e.g. ANN) with the same database and input variables. It can be simply done by integrating another machine learning algorithm on the presented database and input variables in this paper.

Reviewer #2: the citation in text must be check and corrected. There are wrong citation in current version. ex. Mustapha and Mohamed [1] additionally, structure [2], [3]. this one must be [2-3]

What is new in this study should be clearly stated.

limitation of study, parameters used

the parameters which was not considered in this work should be mentioned

"Generally, the role of FA or silica fume is more pronounced at the later age of concrete." why

more discussion should be provided for better understanding

6. PLOS authors have the option to publish the peer review history of their article (what does this mean?). If published, this will include your full peer review and any attached files.

Reviewer #1: No

Reviewer #2: **Yes: **Assoc. Prof. Dr. Ertug Aydin

---

## [Author Response · Author response to Decision Letter 0]

24 Dec 2021

RESPONSES OF THE ACADEMIC EDITOR AND REVIEWER'S COMMENTS

Manuscript: Investigation on Factors Affecting Early Strength of High-Performance Concrete by Gaussian Process Regression

Hai-Bang Ly1,*, Thuy-Anh Nguyen1, Binh Thai Pham1

1University of Transport Technology, Hanoi 100000, Vietnam

*Corresponding authors: Hai-Bang Ly (banglh@utt.edu.vn)

I. ACADEMIC EDITOR

and

Response: 

We thank the academic editor for the comments. We have revised our manuscript in order to meet the journal's rigorous scientific requirements.

 

II. RESPONSE TO REVIEWER #1

General comments: This paper integrates the Gaussian Process Regression (GPR) algorithm for the prediction of early compressive strength of high performance concrete (HPC) using a database including 324 HPC samples showing the influence of input mixture variables on the early strength of HPC. This paper looks interesting and well-written. However, this reviewer recommends the publication of this paper after the following concerns are successfully addressed.

Response:

We thank Reviewer #1 for the encouraging comments. 

Comment #1. There should be more discussion on the effectiveness and applicability of HPC concrete compared to traditional concrete in the Introduction. Please discuss the sustainability of high performance concrete (HPC) in the introduction as it includes industrial wastes such as fly ash and slag making it a greener concrete.

Response:

We thank Reviewer #1 for this comment. Additional discussions are included in the Introduction section of the revised manuscript. All changes are highlighted in red:

"It is vital to emphasize the long-term profitability of the cement sector since the additional cement materials contribute to reducing the quantity of CO2 released throughout the cement production process. Furthermore, since the majority of these compounds are by-products of industrial activities, reusing them is beneficial. As a result, it is predicted that HPC would grow more popular in the next decades, mostly due to its high levels of sustainability and durability." 

Comment #2. Please rephrase "However, in these studies, it was only determined in determining the proportion of material components affecting the compressive strength of concrete without showing the importance of each type of material affecting the compressive strength of HPC". 

Response:

We thank Reviewer #1 for the remark. We have rephrase the sentence as follow:

"However, these works showed the relationship between concrete components that influence the compressive strength without establishing the relevance of each kind of material that affects compressive strength."

Comment #3. There should be more discussion on the advantages (e.g. efficiency with small datasets) and disadvantages (e.g. inefficiency in high dimensional spaces) of Gaussian Process Regression (GPR) algorithm compared to other machine learning algorithms in Section 3.

Response:

We thank Reviewer #1 for the comment. We have added more discussions in the revised manuscript as follow:

“Although the GPR model outperformed the ANN and SVM models in predicting the compressive strength of HPC, it is important to note that each machine learning technique has its own set of benefits and drawbacks that must be considered. First and foremost, the computational requirements for GPR are much higher than those for ANN. As a result, this model can only be employed with small datasets. Furthermore, GPR is better than ANN in dealing with missing data and can properly collect hidden information, even in regions with few available training data points. This is GPR’s primary advantage over other machine learning algorithms and the reason for the PDP analysis in the next sections. Additionally, many different kernels are available for use in the GPR model (standard kernels are supplied in Matlab, for instance), but it is also possible to custom the Kernels. Finally, this method makes the prediction using the whole set of features information. This implies that GPR may become inefficient when the number of features exceeds a few dozens.”

Comment #4. As shown in Figure 3 and Table 1, testing set has a better predictive performance (R = 0.9532) than training set (R = 0.9255) which is not common for machine learning methods where training set has generally a better predictive performance than testing set. Please explain why it happens.

Response:

We thank Reviewer #1 for the comments. In general, k-fold cross-validation (CV) was employed during the training phase of machine learning models, and the prediction performance of the training dataset was defined as the average of k runs. CV with k=5 was utilized exclusively in this investigation (k=5, or k=10 are suggested values to use, according to [1], [2]). CV is used in model selection to guarantee that regardless of the variance in the training data (i.e., sample index in the training and testing datasets), the trained model is always capable of accurately predicting the testing data (not used during the training phase). Thus, the testing set outperforms the training set in terms of predictive performance (R = 0.9532) only owing to the organization (or selection) of the samples in the two datasets. 

Numerous articles on the use of machine learning models in solving civil engineering problems have also shown that the predictive performance of the testing dataset outperform that of the training dataset. For instance, Steel and Composite Structures, Vol. 39, No. 3 (2021) 319-335, (DOI: https://doi.org/10.12989/scs.2021.39.3.319).

Comment #5. In section 4.2, the effectiveness of GPR is shown through a comparison with other machine learning algorithms in the literature with different sample sizes and input variables. However as the predictive performance of machine learning algorithms depends on the input variables and sample size, please show the predictive effectiveness of the presented method compared to another machine learning algorithm (e.g. ANN) with the same database and input variables. It can be simply done by integrating another machine learning algorithm on the presented database and input variables in this paper.

Response:

We thank Reviewer #1 for this suggestion. We have added a short paragraph in the revised manuscript to compare the effectiveness of GPR with ANN and SVM, as follow:

"The predictive performance of GPR was compared with two state-of-the-art machine learning algorithms, namely ANN and SVM. The same training dataset was used to train the two models, whereas the predictive performance was evaluated using the same testing dataset. Regarding the ANN model, one single hidden layer structure was adopted for comparison, with 9 neurons in such hidden layer. The training algorithm was selected as the default training function in Matlab programming language (i.e., Levenberg-Marquardt algorithm), whereas the "tansig" transfer function for the hidden layer and linear function for the output layer, were adopted for comparison. With respect to SVM model, Bayesian optimization was used to find the appropriate hyperparameters for SVM, which is the "fitrsvm" function in Matlab. The final hyperparameters adopted for comparison were BoxConstraint = 19.05, Epsilon = 13.49, Kernel function = polynomial, Kernel polynomial order = 3, Kernel scale = auto, solver = Sequential minimal optimization (SMO). The prediction results are shown in function of all dataset (Fig. 3). It can be seen that although high accuracy was achieved (i.e., RANN = 0.9274, RSVM = 0.9025), the prediction performance of these two models was slightly inferior to the proposed GPR model in this study (RGRP = 0.9329)."

Fig. 4. Regression graphs for the measured and predicted values of early compressive strength of HPC for all dataset: (a) ANN; and (b) SVM.

 

III. RESPONSE TO REVIEWER #2

Comment #1. The citation in text must be check and corrected. There are wrong citation in current version. ex. Mustapha and Mohamed [1] additionally, structure [2], [3]. this one must be [2-3]

Response:

We thank Reviewer #2 for the comments. The reference citation style has been updateded in the revised manuscript. 

Comment #2. What is new in this study should be clearly stated.

Response:

We thank Reviewer #2 for this interesting comment. We have revised our manuscript and added the significance of this study in the introduction section as follow:

"In summary, understanding the parameters that influence the early strength of concrete in general, and high-performance concrete in particular, is critical throughout the design process. The early strength of HPC is also a crucial component to consider, and it should be carefully managed throughout the earliest stages of the construction phase. The early strength is critical since it also dictates the age at which formwork may be removed. "

Comment #3. Limitation of study, parameters used.

Response:

We thank Reviewer #2 for this comment. The limitation of this study is updated in the revised manuscript. Precisely, the database should be enriched and considering more cement replacement materials such as silica fume or metakaolin. This short paragraph is added to the revised manuscript: 

"In general, this study may be valuable in aiding engineers in calculating the appropriate content of mixture components to utilize in the manufacturing process of HPC. Not to mention, cement replacement materials such as silica fume or metakaolin have been shown to be successful in substituting cement while also significantly increasing the mechanical properties of HPC. This has the potential to be a flourishing research topic with applications in a variety of fields if properly developed. Due to the fact that these two components were not included in this study, the primary focus of future research will be to expand the present database and analyze the influence of these factors on the mechanical behavior of HPC."

Also, the parameters used for ML simulation are added to the revised manuscript.

Comment #4. The parameters which was not considered in this work should be mentioned.

Response:

We thank Reviewer #2 for the comments. The cement replacement materials such as silica fume or metakaolin were not considered in this study. We have revised our manuscript and mentioned that these two parameters were not considered in the database:

"It is worth mentioning that other cement replacement materials, such as silica fume or metakaolin, were not included in the current database."

Besides, the conclusion section was also changed with the corresponding perspective (Please also refer to the previous comment).

Comment #5. "Generally, the role of FA or silica fume is more pronounced at the later age of concrete." Why, more discussion should be provided for better understanding

We thank Reviewer #2 for the comments. More discussions are provided in the revised manuscript as follow:

“Indeed, the advantage of HPC is that it often incorporates pozzolanic or latent hydraulic additional components, such as fly ash, silica fume and GGBS. These elements react with (or activated by) alkali created by the hydration of cement to form specific compounds that improve the strength. Precisely, the basic benefit of fly ash is its reactivity with the available lime and alkali in concrete, creating more and more cementitious compounds over time. The pozzolanic reaction of fly ash with lime gives an additional calcium silicate hydrate binder (C-S-H). Besides, silica fume with its fineness at approximately two orders of magnitude finer than the others, has extremely large surface area. Because silica fume is so reactive, it consumes the alkali that the cement eventually releases. Besides, the findings in this study are also in good agreement with the literature [3].”

 

References

[1] Y. Jung and J. Hu, “AK-fold averaging cross-validation procedure,” Journal of nonparametric statistics, vol. 27, no. 2, pp. 167–179, 2015.

[2] B. G. Marcot and A. M. Hanea, “What is an optimal value of k in k-fold cross-validation in discrete Bayesian network analysis?,” Computational Statistics, vol. 36, no. 3, pp. 2009–2031, 2021.

[3] M. Pala, E. Özbay, A. Öztaş, and M. I. Yuce, “Appraisal of long-term effects of fly ash and silica fume on compressive strength of concrete by neural networks,” Construction and Building Materials, vol. 21, no. 2, pp. 384–394, 2007.

---

## [Decision Letter · Decision Letter 1]

10 Jan 2022

Investigation on Factors Affecting Early Strength of High-Performance Concrete by Gaussian Process Regression

PONE-D-21-25655R1

Dear Dr. LY,

We’re pleased to inform you that your manuscript has been judged scientifically suitable for publication and will be formally accepted for publication once it meets all outstanding technical requirements.

Kind regards,

Tianyu Xie, Ph.D.

Academic Editor

PLOS ONE

Additional Editor Comments (optional):

Reviewers' comments:

Reviewer's Responses to Questions

**Comments to the Author**

1. If the authors have adequately addressed your comments raised in a previous round of review and you feel that this manuscript is now acceptable for publication, you may indicate that here to bypass the “Comments to the Author” section, enter your conflict of interest statement in the “Confidential to Editor” section, and submit your "Accept" recommendation.

Reviewer #1: All comments have been addressed

Reviewer #2: All comments have been addressed

2. Is the manuscript technically sound, and do the data support the conclusions?

Reviewer #1: Yes

Reviewer #2: Yes

3. Has the statistical analysis been performed appropriately and rigorously? 

Reviewer #1: Yes

Reviewer #2: Yes

4. Have the authors made all data underlying the findings in their manuscript fully available?

Reviewer #1: Yes

Reviewer #2: Yes

5. Is the manuscript presented in an intelligible fashion and written in standard English?

Reviewer #1: Yes

Reviewer #2: Yes

6. Review Comments to the Author

Reviewer #1: All comments have been addressed in the current version of manuscript and this manuscript can now be accepted

Reviewer #2: all the necessary suggestions and comments are implemented in revised manuscript. In my previous comments are all addressed

7. PLOS authors have the option to publish the peer review history of their article (what does this mean?). If published, this will include your full peer review and any attached files.

Reviewer #1: No

Reviewer #2: **Yes: **Ertug Aydın

---

## [Editor Report · Acceptance letter]

17 Jan 2022

PONE-D-21-25655R1 

Investigation on Factors Affecting Early Strength of High-Performance Concrete by Gaussian Process Regression 

Dear Dr. Ly:

I'm pleased to inform you that your manuscript has been deemed suitable for publication in PLOS ONE. Congratulations! Your manuscript is now with our production department. 

Kind regards, 

on behalf of

Dr. Tianyu Xie 

Academic Editor

PLOS ONE